# The paranodal cytoskeleton clusters Na$^+$ channels at nodes of Ranvier

Veronique Amor[1][†], Chuansheng Zhang[2][†], Anna Vainshtein[1], Ao Zhang[3], Daniel R Zollinger[2], Yael Eshed-Eisenbach[1], Peter J Brophy[3], Matthew N Rasband[2]*, Elior Peles[1]*

[1]Department of Molecular Cell Biology, Weizmann Institute of Science, Rehovot, Israel; [2]Department of Neuroscience, Baylor College of Medicine, Houston, United States; [3]Centre for Neuroregeneration, University of Edinburgh, Edinburgh, United Kingdom

**Abstract** A high density of Na$^+$ channels at nodes of Ranvier is necessary for rapid and efficient action potential propagation in myelinated axons. Na+ channel clustering is thought to depend on two axonal cell adhesion molecules that mediate interactions between the axon and myelinating glia at the nodal gap (i.e., NF186) and the paranodal junction (i.e., Caspr). Here we show that while Na$^+$ channels cluster at nodes in the absence of NF186, they fail to do so in double conditional knockout mice lacking both NF186 and the paranodal cell adhesion molecule Caspr, demonstrating that a paranodal junction-dependent mechanism can cluster Na$^+$ channels at nodes. Furthermore, we show that paranode-dependent clustering of nodal Na$^+$ channels requires axonal $\beta$II spectrin which is concentrated at paranodes. Our results reveal that the paranodal junction-dependent mechanism of Na$^+$channel clustering is mediated by the spectrin-based paranodal axonal cytoskeleton.

*For correspondence: rasband@ bcm.edu (MNR); peles@weizmann. ac.il (EP)

[†]These authors contributed equally to this work

Competing interests: The authors declare that no competing interests exist.

## Introduction

Nodes of Ranvier are short gaps in the myelin sheath where voltage-gated Na$^+$ and K$^+$ channels are clustered in high densities to facilitate the rapid and efficient propagation of action potentials. Node assembly depends on the interactions between axons and myelinating glia that are mediated by distinct cell adhesion complexes present at both the nodal gap and the paranodal junction flanking each node of Ranvier. A key mediator of these interactions is the cell adhesion molecule neurofascin (Nfasc). The *Nfasc* gene is alternatively spliced to generate a neuronal 186 kDa variant (NF186) found at nodes of Ranvier (*Davis et al., 1996*), and a glial 155 kDa variant (NF155) that is a core component of the paranodal junctions (*Tait et al., 2000*; *Pillai et al., 2009*). *Nfasc* is essential for node of Ranvier assembly as genetic deletion of both NF155 and NF186 in mice results in the absence of nodal Na$^+$ channel clusters (*Sherman et al., 2005*).

At peripheral (PNS) and central (CNS) nodes, NF186 is clustered and stabilized at the axolemma by glial adhesion molecules and extracellular matrix (ECM) (*Eshed et al., 2005*; *Feinberg et al., 2010*; *Susuki et al., 2013*; *Colombelli et al., 2015*). Clustered NF186 functions as an attachment site for the nodal scaffolding proteins ankyrinG (AnkG) and $\beta$IV spectrin, which in turn recruit Na$^+$ channels (*Davis and Bennett, 1994*; *Sherman et al., 2005*; *Yang et al., 2007*; *Gasser et al., 2012*; *Ho et al., 2014*). After node assembly, NF186 helps to stabilize and maintain the nodal Na$^+$ channel protein complex (*Amor et al., 2014*; *Desmazieres et al., 2014*). Consistent with these observations, transgenic expression of neuronal NF186 on an *Nfasc*-null background is sufficient to rescue nodal Na$^+$ channel clustering (*Zonta et al., 2008*). In addition, the glial NF155-dependent paranodal axo-glial junctions have also been proposed to function as a parallel mechanism to cluster Na$^+$ channels

by acting as lateral diffusion barriers to restrict the location of nodal proteins in the axolemma (*Rasband et al., 1999*; *Pedraza et al., 2001*). In support of this idea, transgenic expression of NF155 in myelinating oligodendrocytes on a *Nfasc*-null background is sufficient to induce Na$^+$ channel clustering (*Zonta et al., 2008*). Similarly, in vitro myelination of *Nfasc*-null axons by wild-type Schwann cells (which results in the formation of a normal paranodal junction) induces the clustering of Na$^+$ channels at nodes (*Feinberg et al., 2010*). Furthermore, mice lacking both paranodal junctions and NF186-binding CNS nodal ECM proteins have a profound reduction in the clustering of Na$^+$ channels (*Susuki et al., 2013*). In contrast, disruption of paranodal junctions alone causes only mild perturbations to Na$^+$ channel clustering (*Dupree et al., 1999*; *Bhat et al., 2001*; *Boyle et al., 2001*; *Pillai et al., 2009*). Together, these observations support a model where two glia-dependent mechanisms direct Na$^+$ channel clustering at nodes of Ranvier: (1) clustering of axonal NF186 by glia-derived proteins and (2) restriction of nodal protein complexes within the nodal gap by the paranodal junctions.

Nevertheless, this model, and in particular the function of the paranodal junction in the clustering of Na$^+$ channels, remains controversial since conditional deletion of NF186 in neurons was reported to block the clustering of Na$^+$ channels at nodes of Ranvier (*Thaxton et al., 2011*). Furthermore, if paranodal junctions can act as diffusion barriers to cluster Na$^+$ channels, what molecular mechanisms are involved?

To further determine if paranodal junctions are sufficient to cluster nodal Na$^+$ channels in vivo, we genetically deleted (in three separate laboratories) nodal NF186 using two different conditional *Nfasc* alleles, and three independent Cre-driver lines. We confirmed the role of the paranodal junction as a second, glia-dependent mechanism for Na$^+$ channel clustering by generating double-conditional knockout mice lacking both axonal Caspr and NF186. Finally, using double-conditional knockout mice deficient in both axonal NF186 and $\beta$II spectrin, we extend our understanding of node formation by showing that the axonal $\beta$II spectrin-based paranodal cytoskeleton underlies the paranodal mechanism of nodal Na$^+$ channel clustering.

## Results

### NF155-dependent paranodal junctions can cluster Na$^+$ channels in the PNS

To determine whether the paranodal junctions are sufficient to cluster Na$^+$ channels in the absence of axoglial contact at the nodes, we generated two distinct *Nfasc* conditional alleles to specifically remove NF186 in neurons: mice with floxed *Nfasc* exons 6 and 7, and mice with a floxed *Nfasc* exon 4, hereafter referred to as *Nfasc$^{fl/fl}$* and *Nfasc(4)$^{fl/fl}$*, respectively (*Figure 1—figure supplement 1a* and *Zonta et al., 2011*). We confirmed the efficiency of the targeting strategy by crossing *Nfasc$^{fl/fl}$* mice with *Pgk-Cre* mice (*Pgk-Cre* results in recombination in all tissues). Immunoblots of brain lysates derived from postnatal day 6 (P6) *Pgk-Cre;Nfasc$^{fl/fl}$* mice showed no immunoreactivity for *Nfasc* gene products (*Figure 1—figure supplement 1b*). Consistent with the previously reported *Nfasc*-null mice (*Sherman et al., 2005*), *Pgk-Cre;Nfasc$^{fl/fl}$* mice died within a few days of birth and immunostaining of sciatic nerves showed neither paranodal junctions nor Na$^+$ channel clustering (*Figure 1—figure supplement 1c and d*). Crossing *Nfasc(4)$^{fl/fl}$* mice with *Thy1-Cre* mice (*Thy1Cre* mice express Cre recombinase in neurons) also resulted in loss of NF186 in *Thy1-Cre;Nfasc(4)$^{fl/fl}$* mice (*Figure 1—figure supplement 2a*).

To avoid deletion of the glial NF155 form of neurofascin and disruption of the paranodal junctions, we crossed *Nfasc$^{fl/fl}$* mice with *Avil-Cre* mice (*Hasegawa et al., 2007*; *Zhou et al., 2010*), which restricts the Cre-mediated recombination to PNS sensory neurons beginning at E12.5. Immunostaining of dorsal roots from *Avil-Cre;Nfasc$^{fl/fl}$* mice using a neurofascin antibody that recognizes all splice variants (panNF) showed complete loss of nodal NF186, but preserved paranodal junctions as indicated by Caspr and paranodal panNF immunostaining (*Figure 1a*); ventral roots were unaffected and the mixed fiber type sciatic nerve showed some nodes with NF186 and others without (*Figure 1a*). To determine if the paranodal junctions are intact in *Avil-Cre;Nfasc$^{fl/fl}$* mice we performed transmission electron microscopy on dorsal roots and found no disruption of the junctions (*Figure 1—figure supplement 1e*). Thus, nodes of Ranvier corresponding to sensory axons in *Avil-Cre;Nfasc$^{fl/fl}$* mice lack NF186 and their paranodal junctions are intact.

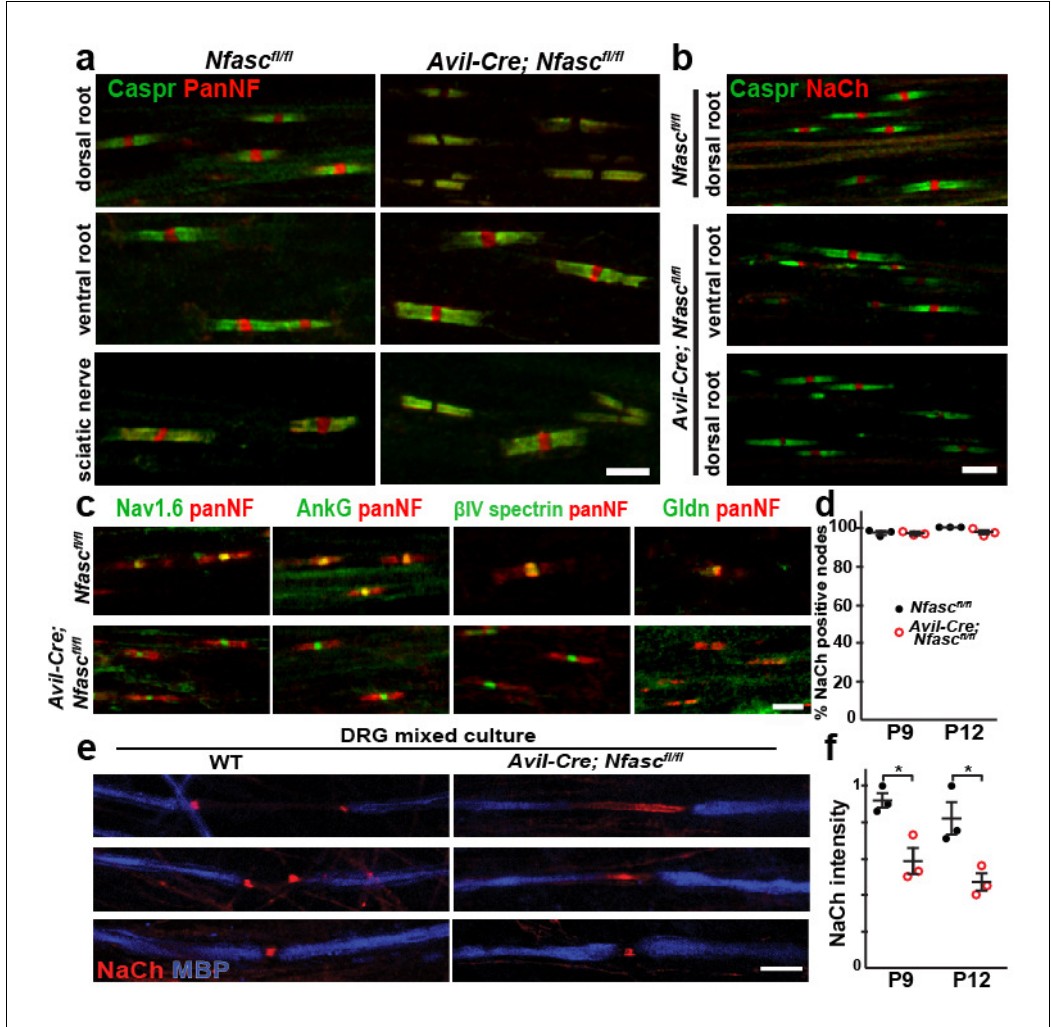

**Figure 1.** NF155-dependent paranodal junctions can cluster Na$^+$ channels in the PNS. (a) Immunostaining of P12 dorsal roots, ventral roots, and sciatic nerve from *Nfasc$^{fl/fl}$* and *Avil-Cre;Nfasc$^{fl/fl}$* mice using a pan Nfasc (Pan-NF, red) and Caspr (green) antibodies. Note the yellow overlap of the two labels at the paranodes in both genotypes. Scale bar, 5 μm. (b) Immunostaining of P12 dorsal and ventral roots from *Nfasc$^{fl/fl}$* and *Avil-Cre;Nfasc$^{fl/fl}$* mice using pan Na$^+$ channel (NaCh, red) and Caspr (green) antibodies. Scale bar, 5 μm. (c) Immunostaining of P12 dorsal roots from *Nfasc$^{fl/fl}$* and *Avil-Cre;Nfasc$^{fl/fl}$* mice using panNF (red) to label nodes and paranodes, and antibodies against Nav1.6, AnkG, βIV spectrin, or Gldn to label nodes (green). Note that the nodal panNF staining observed in the *Nfasc$^{fl/fl}$* mice is much stronger than the paranodal immunoreactivity. Thus, the paranodal panNF staining is more pronounced in the *Avil-Cre;Nfasc$^{fl/fl}$* mice. Scale bar, 5 μm. (d) Quantification of the percentage of dorsal root nodes of Ranvier with nodal Na$^+$ channels at P9 and P12. N = 3 mice at each time point for each genotype. (e) Immunostaining of dorsal root ganglion/Schwann cell mixed cultures from wild-type and *Avil-Cre;Nfasc$^{fl/fl}$* mice using antibodies against NaCh (red) and myelin basic protein (MBP, blue). Scale bar = 10 μm. (f) Quantification of the relative Na$^+$ channel fluorescence intensity at P9 and P12. N = 3 mice at each time point for each genotype. *p=0.016 at P9; p=0.026 at P12. The number of nodes measured at P9 were 263 and 250, at P12 were 337 and 269 in the *Nfasc$^{fl/fl}$* and *Avil-Cre;Nfasc$^{fl/fl}$* dorsal roots, respectively.

The following figure supplements are available for figure 1:

**Figure supplement 1.** Characterization of *Pgk-Cre;Nfasc$^{fl/fl}$* and *Avil-Cre;Nfasc$^{fl/fl}$* mice.

**Figure supplement 2.** NF155-dependent paranodal junctions can cluster Na$^+$ channels in the PNS and CNS.

To determine if the paranodal junctions can cluster nodal Na$^+$ channels in the absence of axonal NF186, we examined dorsal roots from both *Nfasc$^{fl/fl}$* and *Avil-Cre;Nfasc$^{fl/fl}$* mice by immunostaining. We found clusters of Na$^+$ channels at all nodes of Ranvier at both P9 and P12 (*Figure 1b–d*). We also examined the clustering of other nodal proteins including AnkG, βIV spectrin, and gliomedin (Gldn; *Figure 1c*). Whereas Na$^+$ channels, AnkG, and βIV spectrin were unaffected by the loss of NF186, Gldn was no longer restricted to nodes of Ranvier, consistent with gliomedin's function as a ligand for NF186 (*Eshed et al., 2005*). Similar to *Avil-Cre;Nfasc$^{fl/fl}$* mice, analysis of peripheral nerves from *Thy1-Cre;Nfasc(4)$^{fl/fl}$* mice revealed clustered Na$^+$ channels, AnkG, and βIV spectrin, but loss of the nodal NF186 ligands NrCAM and Gldn (*Figure 1—figure supplement 2b and d*).

A major function of NF186 during early development is to interact with gliomedin to form hemi-nodal Na$^+$ channel clusters which are precursors to mature nodal Na$^+$ channel clusters (*Feinberg et al., 2010*). Heminodes are most easily seen in dorsal root ganglion/Schwann cell mixed cultures that have been induced to myelinate. Whereas 84% (n = 128 heminodes) of heminodes from wild-type (WT) cultures had heminodal Na$^+$ channel clusters (*Figure 1e*), only 6% (n = 113 heminodes) of heminodes from *Avil-Cre;Nfasc$^{fl/fl}$* mice had heminodal Na$^+$ channel clusters. Instead, *Avil-Cre;Nfasc$^{fl/fl}$* mice had Na$^+$ channels in the forming nodal gap between adjacent Schwann cells (*Figure 1e*). The distribution of Na$^+$ channels in the latter supports a role for paranodal junctions in restricting nodal proteins between two forming myelin sheaths (*Feinberg et al., 2010*). Intriguingly, close examination of *Avil-Cre;Nfasc$^{fl/fl}$* dorsal roots and *Thy1-Cre;Nfasc(4)$^{fl/fl}$* peripheral nerves showed that the intensity of nodal Na$^+$ channel staining was reduced (*Figure 1b* and *Figure 1—figure supplement 2b*). Quantification of the nodal Na$^+$ channel fluorescence intensity in *Avil-Cre; Nfasc$^{fl/fl}$* mice at P9 and P12 showed a significant reduction compared to control *Nfasc$^{fl/fl}$* mice (*Figure 1f*). Consistent with this reduced nodal Na$^+$ channel density, compound action potential recordings from P10 dorsal roots (*Figure 1—figure supplement 1f*) revealed a reduction in conduction velocity in *Avil-Cre;Nfasc$^{fl/fl}$* mice (*Figure 1—figure supplement 1g*). The loss of Na$^+$ channels from nodes may account for the juvenile lethality observed in *Avil-Cre;Nfasc$^{fl/fl}$* mice (i.e., at 7–10 days these mice began to develop a progressive tremor that worsened until most mice died at around 3 weeks of age). Hence, while these observations confirm that paranodal junctions are sufficient for the initial Na$^+$ channel clustering at PNS nodes, they also support previous observations that NF186 plays important roles in maintaining the nodal protein complex which is necessary for proper action potential conduction (*Amor et al., 2014*; *Desmazieres et al., 2014*).

## The paranodal junctions can cluster Na$^+$ channels in the CNS

To determine if paranodal junctions can cluster Na$^+$ channels in the absence of axonal NF186 in the CNS, we crossed *Nfasc$^{fl/fl}$* mice with *Six3-Cre* mice which are reported to undergo recombination in retinal ganglion cells (*Furuta et al., 2000*). Immunostaining of *Nfasc$^{fl/fl}$* mouse optic nerves with a chicken panNF antibody that recognizes both NF186 and NF155 splice variants showed staining at both nodes and paranodes (note the yellow color at nodes due to overlap with Na$^+$ channels in the *Nfasc$^{fl/fl}$* control optic nerves, *Figure 2a*). However, we found that some oligodendrocytes in the *Six3-Cre;Nfasc$^{fl/fl}$* mouse also recombine, resulting in a mosaic nerve with some regions lacking both NF155 and NF186 (*Figure 2—figure supplement 1a–b*, outlined region). Like *Pgk-Cre;Nfasc$^{fl/fl}$* mice, regions lacking both NF155 and NF186 had dramatically reduced Na$^+$ channel clustering; when Na$^+$ channels were clustered in regions lacking paranodal NF155, they always colocalized with NF186 in axons corresponding to retinal ganglion cells that presumably did not undergo recombination (*Figure 2—figure supplement 1b*). Therefore, we restricted our analysis to regions of the *Six3-Cre;Nfasc$^{fl/fl}$* mouse optic nerve where oligodendrocytes still expressed NF155. Immunostaining showed a pronounced gap in Nfasc immunoreactivity at nodes; occasionally we found some nodes that still had nodal NF186, likely due to a lack of recombination in those neurons (*Figure 2a and b*, arrows). Consistent with our observations in the PNS, we found robust Na$^+$ channel clustering in the gap between all panNF-labeled paranodal junctions at P17 and P30 in the *Six3-Cre;Nfasc$^{fl/fl}$* mouse optic nerve (*Figure 2a,b,d*). Intriguingly, at P60 some nodes had intact panNF-labeled paranodal junctions but without any associated Na$^+$ channel immunostaining (*Figure 2c*, arrowhead and *Figure 2d*). However, in contrast to the PNS (*Figure 1f*), we did not observe a general reduction in Na$^+$ channel fluorescence intensities (*Figure 2e*). As expected, nodes lacking NF186 in the *Six3-Cre; Nfasc$^{fl/fl}$* mice had AnkG and βIV spectrin (*Figure 2—figure supplement 1c*). Furthermore, we did not observe any invasion of Kv1.2-subunit containing K$^+$ channels into paranodal regions, indicating

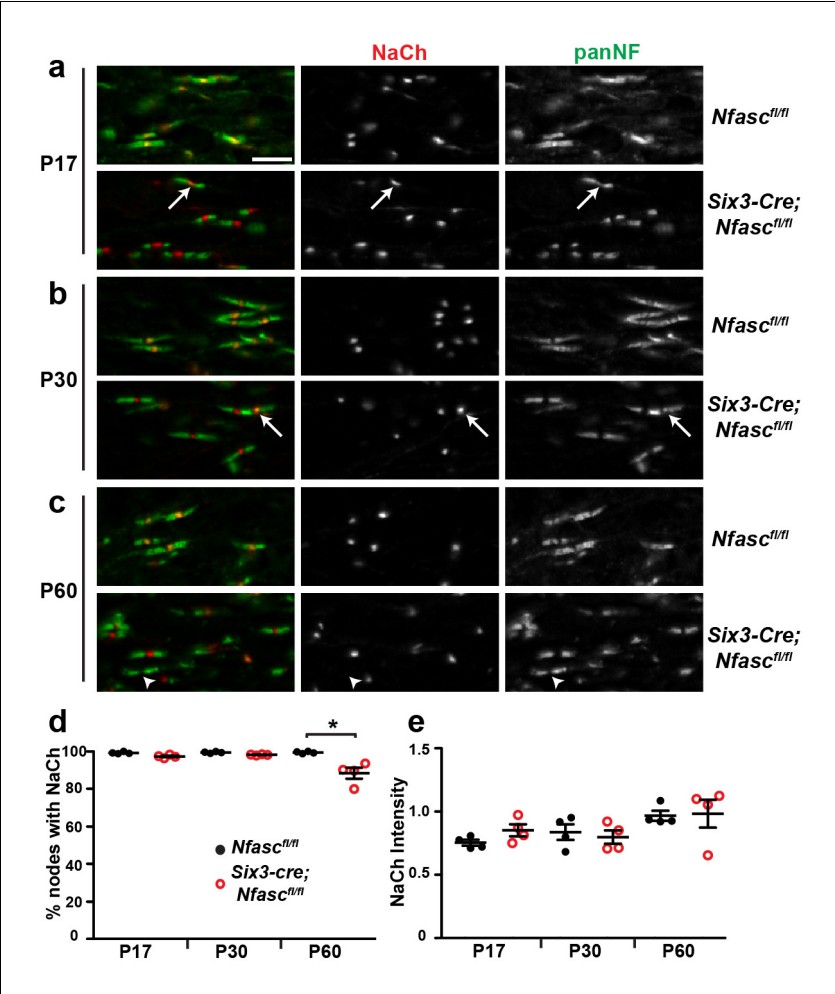

**Figure 2.** NF155-dependent paranodal junctions can cluster Na$^+$ channels in the CNS. (a–c) Immunostaining of P17, P30, and P60 optic nerves from *Nfasc$^{fl/fl}$* and *Six3-Cre;Nfasc$^{fl/fl}$* mice using chicken panNF (green) and mouse monoclonal pan Na$^+$ channel (NaCh, red) antibodies. Scale bar, 10 μm. (d) The percentage of panNF nodes that have Na$^+$ channels. N = 4 animals per time point per genotype. *p=0.03. (e) Nodal Na$^+$ channel immunofluorescence intensity at P17, P30, and P60 in optic nerves from *Nfasc$^{fl/fl}$* and *Six3-Cre;Nfasc$^{fl/fl}$* mice. N = 4 animals per time point per genotype.

The following figure supplement is available for figure 2:

**Figure supplement 1.** Characterization of *Six3-Cre;Nfasc$^{fl/fl}$* mice.

that the paranodal axoglial junctions remain intact in the *Six3-Cre;Nfasc$^{fl/fl}$* mouse optic nerve (*Figure 2—figure supplement 1c*). Similarly, analysis of nodes of Ranvier in the spinal cords of *Thy1-Cre;Nfasc(4)$^{fl/fl}$* showed that despite the loss of NF186, there was robust clustering of Na$^+$ channels, AnkG, and βIV spectrin (*Figure 1—figure supplement 2c and d*). However, the chondroitin-sulfate proteoglycan brevican (Bcan), a core nodal ECM protein that binds to NF186 (*Susuki et al., 2013*), could not be detected (*Figure 1—figure supplement 2c*).

Similar to our observations in the PNS, compound action potential recordings from *Nfasc$^{fl/fl}$* and *Six3-Cre;Nfasc$^{fl/fl}$* mouse optic nerves showed a significant reduction in conduction velocity at all time points analyzed (*Figure 2—figure supplement 1d and 1e*). The reduced conduction velocity does not reflect impaired myelination (*Figure 2—figure supplement 1f and 1g*), but instead likely reflects the combined effects of decreased nodal Na$^+$ channel density, loss of nodal extracellular matrix molecules (*Weber et al., 1999*; *Bekku et al., 2010*), and loss of some paranodal junctions

due to NF155-deficient oligodendrocytes (*Figure 2—figure supplement 1b*). Together, these observations reinforce the concept that paranodal junctions in CNS myelinated axons are sufficient to induce Na$^+$ channel clustering in the absence of NF186. Furthermore, they extend our understanding of CNS nodes by showing that assembly of the CNS nodal ECM depends on NF186, and that as in the PNS, NF186 may contribute to the maintenance of nodal Na$^+$ channel densities required for proper action potential propagation.

## Paranodal junctions constitute a second mechanism for nodal Na$^+$ channel clustering

How are Na$^+$ channels clustered in the absence of NF186? The prevailing evidence points to a model where the paranodal junctions support the clustering of Na$^+$ channels (*Zonta et al., 2008*; *Feinberg et al., 2010*; *Susuki et al., 2013*). To extend these previous studies and to directly test this hypothesis we generated double-conditional knockout mice lacking both Caspr and NF186 in sensory neurons: *Avil-Cre;Nfasc$^{fl/fl}$;Caspr$^{fl/fl}$*. In control ventral roots NF186 is found at nodes, Caspr and NF155 at paranodes, and Kv1.1 K$^+$ channels at juxtaparanodes (*Figure 3a*). Immunostaining of dorsal roots from *Avil-Cre;Nfasc$^{fl/fl}$;Caspr$^{fl/fl}$* mice showed complete loss of Caspr-labeled paranodal junctions and nodal NF186, as well as a redistribution of Kv1.1 K$^+$ channels into paranodal regions (*Figure 3b*, arrowheads). Furthermore, immunostaining for Nav1.6 Na$^+$ channels or $\beta$IV spectrin showed a complete loss of clustering of these nodal proteins (*Figure 3c,d*). Thus, loss of both NF186 and the paranodal axoglial junctions blocks the assembly of nodes, confirming that intact paranodes function as a second, independent mechanism for Na$^+$ channel clustering.

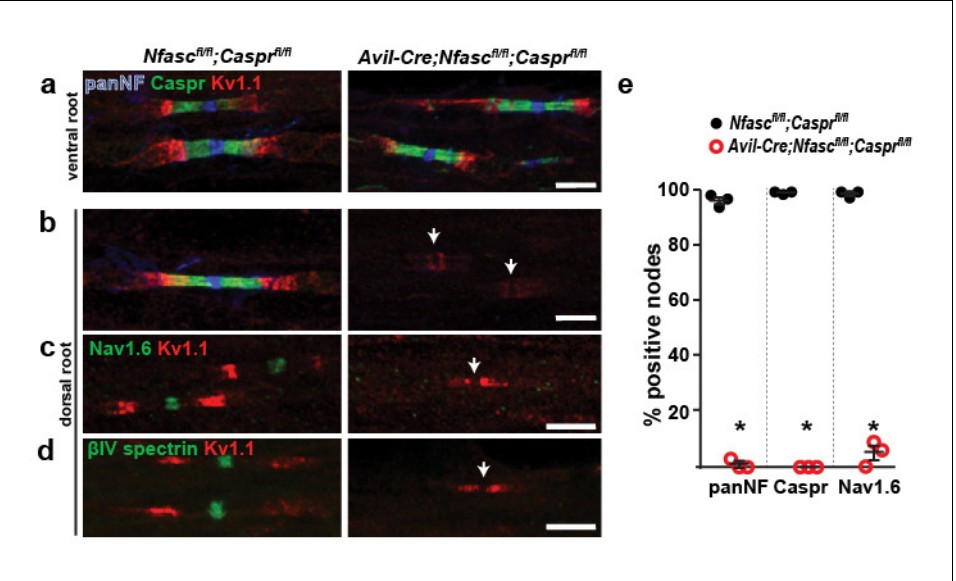

**Figure 3.** Paranodes can cluster Na$^+$ channels at nodes of Ranvier. (a, b) Immunostaining of P14 *Nfasc$^{fl/fl}$;Caspr$^{fl/fl}$* and *Avil-Cre;Nfasc$^{fl/fl}$;Caspr$^{fl/fl}$* ventral (a) and dorsal (b) roots using antibodies against panNF (blue), Caspr (green), and Kv1.1 (red). Arrows in (b) indicate the location of nodes. Scale bars, 10 μm. (c, d) Immunostaining of P14 *Nfasc$^{fl/fl}$;Caspr$^{fl/fl}$* and *Avil-Cre;Nfasc$^{fl/fl}$;Caspr$^{fl/fl}$* dorsal roots using antibodies against Nav1.6 Na$^+$ channels (c, green), $\beta$IV spectrin (d, green), and Kv1.1 (red). Arrows indicate the location of nodes. Scale bars, 10 μm. (e) The percentage of nodes or paranodes labeled with the indicated antibodies in P14 *Nfasc$^{fl/fl}$;Caspr$^{fl/fl}$* and *Avil-Cre; Nfasc$^{fl/fl}$;Caspr$^{fl/fl}$* dorsal roots. N = 3 animals per genotype. Number of nodes analyzed in *Nfasc$^{fl/fl}$;Caspr$^{fl/fl}$* dorsal root: PanNF, 288; Caspr, 298; Nav1.6, 301. Number of nodes analyzed in *Avil-Cre;Nfasc$^{fl/fl}$;Caspr$^{fl/fl}$* dorsal root: PanNF, 365; Caspr, 365; Nav1.6, 336. *PanNF, p=0.0002; Caspr, p=8.07E-06; Nav1.6, p=0.0012.

# The spectrin-based paranodal cytoskeleton underlies the paranodal clustering mechanism

How do paranodal junctions act as a barrier to restrict nodal proteins to the gaps in the myelin sheath? We previously showed that the enrichment of AnkG in the axon initial segment (AIS) depends on an αII/βII spectrin-dependent submembranous cytoskeleton that functions as an intra-axonal boundary (*Galiano et al., 2012*). Paranodes also have a specialized αII/βII spectrin-based axonal cytoskeleton whose assembly requires the NF155-dependent paranodal junction (*Ogawa et al., 2006*). Thus, the paranodal cytoskeleton may function as repeating intra-axonal boundaries that restrict AnkG and Na$^+$ channels to nodes. To test this hypothesis, we generated *Avil-Cre;Nfasc$^{fl/fl}$; Sptbn1$^{fl/fl}$* mice that lack both NF186 and βII spectrin in PNS sensory neurons. Importantly, 94.1% and 94.6% of nodes in control and *Avil-Cre; Sptbn1$^{fl/fl}$* mice, respectively, had clustered Na$^+$ channels (p=0.28 by Student's t-test; n = 3 independent animals for each genotype and a total of 796 and 464 nodes were examined in control and cKO mice, respectively). Furthermore, at individual nodes in the *Avil-Cre;Sptbn1$^{fl/fl}$* mice we found Na$^+$ channel fluorescence intensities of 1.035 and 0.935 (arbitrary units) in control and *Avil-Cre;Sptbn1$^{fl/fl}$* mice, respectively (p=0.44 by Student's t-test; n = 4 mice of each genotype with a total of 100 and 92 nodes measured in control and cKO mice, respectively). Thus, *Avil-Cre;Sptbn1$^{fl/fl}$* mice have normal Na$^+$ channel clustering and densities of Na$^+$ channels, indicating that NF186 is sufficient to cluster nodal Na$^+$ channels. Compared to single knockouts (*Figure 1* and (*Zhang et al., 2013*) and *Nfasc$^{fl/f}$;Sptbn1$^{fl/fl}$* mice), animals lacking both NF186 and βII spectrin in their sensory axons had an extreme deficit in proprioception (*Video 1*). Electron microscopy showed paranodes, but thinner myelin in *Avil-Cre;Nfasc$^{fl/fl}$;Sptbn1$^{fl/fl}$* mice (*Figure 4a–d*), whereas mice lacking βII spectrin had normal myelination (*Zhang et al., 2013*). To confirm that paranodal junctions are intact we immunostained paranodes in dorsal roots using antibodies to Caspr. Both *Nfasc$^{fl/fl}$* and *Avil-Cre;Nfasc$^{fl/fl}$;Sptbn1$^{fl/fl}$* mice had normal paranodal Caspr (*Figure 4e*), indicating intact paranodal junctions. Nevertheless, immunostaining of roots lacking NF186 and βII spectrin revealed a profound loss of Na$^+$ channel, AnkG, and βIV spectrin clustering (*Figure 4F–I*) at P10 and P15 despite intact NF155-dependent paranodal junctions. These results extend the model for how Na$^+$ channels are clustered at nodes by demonstrating that the paranodal junction barrier mechanism is mediated by the βII spectrin-based paranodal cytoskeleton.

## Discussion

We used conditional knockout mice to determine the contributions of paranodal junctions and the paranodal spectrin-based cytoskeleton to Na$^+$ channel clustering at nodes of Ranvier. Our results confirm and extend the model that two glia-dependent mechanisms direct Na$^+$ channel clustering at nodes of Ranvier (*Feinberg et al., 2010*; *Susuki et al., 2013*). These mechanisms consist of: (1) glia-derived adhesion/ECM proteins working through NF186, and (2) the assembly of a paranodal junction dependent cytoskeletal barrier containing βII spectrin (*Figure 5A*). These two mechanisms converge on the clustering of the axonal cytoskeletal scaffolding proteins AnkG and βIV spectrin at the nodes of Ranvier (*Ho et al., 2014*). Accordingly, the loss of NF186 can be partially compensated for by the paranodal junction-based cytoskeletal barrier since Na$^+$ channels may still be clustered, but they cannot be maintained in the absence of NF186 (*Figure 5B*). With the same rationale, Na$^+$ channels still cluster at nodes in the absence of the paranodal junction barrier (by genetically deleting cell adhesion molecules that mediate axoglial contact at this site (*Bhat et al., 2001*; *Boyle et al., 2001*; *Pillai et al., 2009*), or βII spectrin [*Zhang et al., 2013*]) due to the function of NF186 at the forming nodes (*Figure 5C*). When both NF186 and the paranodal junction based spectrin cytoskeleton are lost, Na$^+$ channels cannot be clustered at nodes (*Figure 5D*).

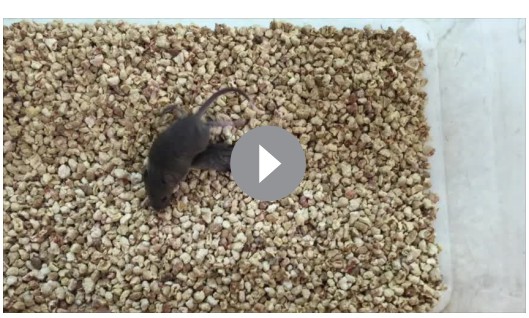

**Video 1.** Movie of P17 *Nfasc$^{fl/fl}$;Sptbn1$^{fl/fl}$* and P17 *Avil-Cre;Nfasc$^{fl/fl}$;Sptbn1$^{fl/fl}$* mice. This video relates to *Figure 4*.

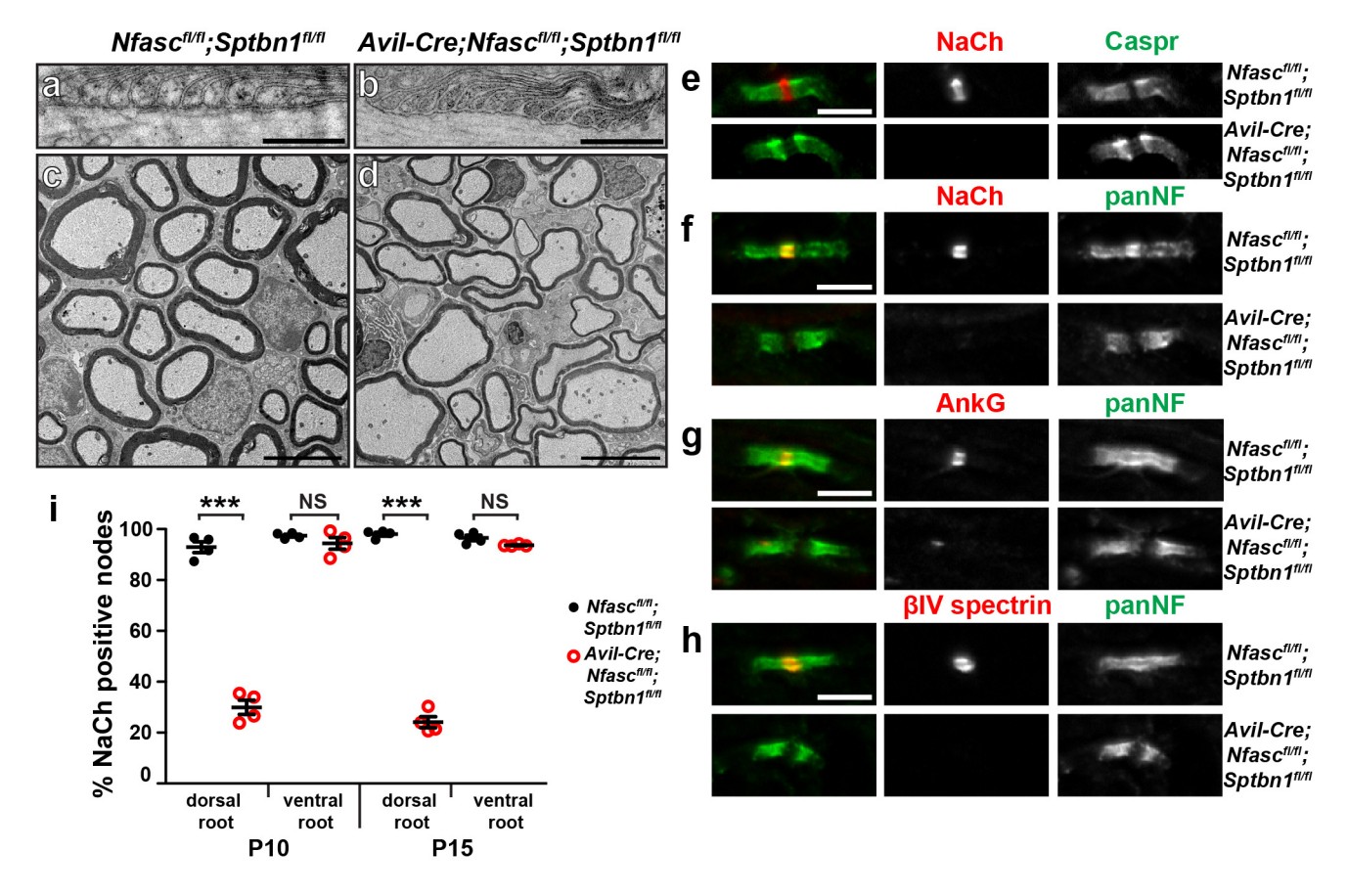

**Figure 4.** The paranodal spectrin-based cytoskeleton can assemble nodes of Ranvier. (a, b) TEM of longitudinal sections from dorsal roots of *Nfasc*<sup>fl/fl</sup>; *Sptbn1*<sup>fl/fl</sup> (a) and *Avil-Cre;Nfasc*<sup>fl/fl</sup>;*Sptbn1*<sup>fl/fl</sup> mice (b). Scale bars, 0.5 μm. (c, d) TEM of cross sections through dorsal roots of *Nfasc*<sup>fl/fl</sup>;*Sptbn1*<sup>fl/fl</sup> (c) and *Avil-Cre;Nfasc*<sup>fl/fl</sup>;*Sptbn1*<sup>fl/fl</sup> mice (d). Scale bars, 4 μm. (e) Immunostaining of P10 *Nfasc*<sup>fl/fl</sup>;*Sptbn1*<sup>fl/fl</sup> and *Avil-Cre;Nfasc*<sup>fl/fl</sup>;*Sptbn1*<sup>fl/fl</sup> dorsal roots using antibodies against $Na^+$ channels and Caspr shows intact paranodal junctions. Scale bar, 5 μm. (f–h) Immunostaining of P15 *Nfasc*<sup>fl/fl</sup>;*Sptbn1*<sup>fl/fl</sup> and *Avil-Cre;Nfasc*<sup>fl/fl</sup>;*Sptbn1*<sup>fl/fl</sup> dorsal roots using antibodies against $Na^+$ channels (e, red), AnkG (f, red), βIV spectrin (g, red) and panNF (green). Scale bar, 5 μm. (i) The percentage of nodes labeled for $Na^+$ channels in P10 and P15 *Nfasc*<sup>fl/fl</sup>;*Sptbn1*<sup>fl/fl</sup> and *Avil-Cre;Nfasc*<sup>fl/fl</sup>;*Sptbn1*<sup>fl/fl</sup> dorsal and ventral roots. N = 4 animals per age and genotype. ***p=2.0E−06 at P10; p=6.26E−08 at P15.

How do glia direct the assembly of the paranodal axonal cytoskeleton? Glial NF155 binds to a heterodimeric cell adhesion molecule complex consisting of axonal Caspr and contactin (*Charles et al., 2002*). In myelinating glia, NF155 also interacts with ankyrinB (AnkB) and AnkG to facilitate the assembly of the paranodal junction, which in turn controls the rapid development of $Na^+$ channel clustering in the CNS (*Chang et al., 2014*). Axonal Caspr has a cytoplasmic protein 4.1 binding domain, and protein 4.1B is enriched at paranodes and juxtaparanodes. Protein 4.1B binds to αII/βII spectrin and mice deficient in either 4.1B or βII spectrin have impaired axonal membrane domain organization with mislocalization of juxtaparanodal Kv1 $K^+$ channels due to loss of the paranodal boundary (*Horresh et al., 2010*; *Einheber et al., 2013*; *Zhang et al., 2013*). Together, these observations suggest that in myelinated axons, glial NF155 directs the assembly of the αII/βII spectrin-based paranodal cytoskeleton through Caspr and protein 4.1B. One prediction from this model is that mice lacking both NF186 and protein 4.1B should phenocopy the *Avil-Cre;Nfasc*<sup>fl/fl</sup>;*Sptbn1*<sup>fl/fl</sup> mice we analyzed.

How does the βII spectrin-based paranodal cytoskeleton function as a barrier to restrict AnkG/βIV spectrin to nodes of Ranvier? We speculate that similar to the AIS and distal axon (*Galiano et al., 2012*), different spectrin and ankyrin cytoskeletons occupy mutually exclusive domains due to steric effects. Thus, domains containing βIV spectrin are mutually exclusive to those

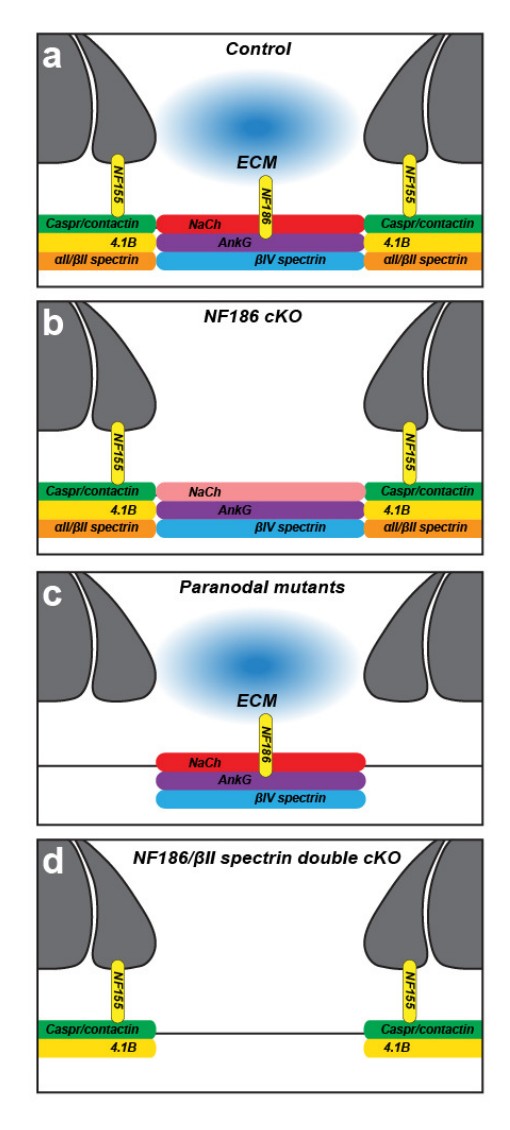

**Figure 5.** Two glia-dependent mechanisms can cluster Na$^+$ channels at nodes of Ranvier. (**a**) Cartoon of control node of Ranvier. (**b**) Na$^+$ channels are clustered in NF186-deficient mice by a paranodal spectrin cytoskeleton. However, the density of Na$^+$ channels is reduced in the PNS. (**c**) Na$^+$ channels are clustered in NF155-deficient and other paranodal mutant mice by NF186-ECM interactions. (**d**) Na$^+$ channels fail to cluster at nodes of Ranvier when both NF186 and the paranodal cytoskeleton are lost.

containing $\beta$II spectrin. Recent experiments using super-resolution microscopy reveal that the $\beta$II spectrin-based cytoskeleton in unmyelinated axons is organized into repeating rings of spectrin tetramers and actin (*Xu et al., 2013*). However, the high density of $\alpha$II/$\beta$II spectrin found at paranodes may not be organized in the same way, and this high density may function to restrict ankG/$\beta$IV spectrin-containing cytoskeletons to the forming nodes as the myelin sheath elongates. While the conclusion that the paranodal junction contributes to the clustering of nodal Na$^+$ channels is consistent with previous studies (*Rasband et al., 1999*; *Zonta et al., 2008*; *Feinberg et al., 2010*), it is different from the one reached by Thaxton et al. (*Thaxton et al., 2011*). To specifically target NF186 these authors used a Cre-recombinase under control of the Neurofilament light chain (*Nefl-Cre*). Unfortunately, this Cre-driver line has poor penetrance and poor temporal precision (*Thaxton et al., 2011*). Since we also observed reduced densities of nodal Na$^+$ channel clustering (in the PNS) and even complete loss of channels in some NF186-deficient axons (both PNS and CNS) with increasing age, we speculate the results of *Thaxton et al. (2011)* reflect the role of NF186 for maintenance of Na$^+$ channel clusters rather than their initial assembly. This conclusion is consistent with their observation that in P11 sciatic nerves, 30% of nodes lacking NF186 still had Na$^+$ channel clustering. It was with these concerns in mind that we used two different *Nfasc* alleles and three different highly penetrant and temporally precise Cre-driver lines. Why are Na$^+$ channel intensities at PNS nodes reduced but normal at CNS nodes in NF186-deficient axons? We speculate that differences in primary and secondary clustering mechanisms can account for the reduced Na$^+$ channel density in the PNS. For example, we previously proposed the main clustering mechanism in the PNS is through ECM/NF186 rather than the paranode (*Feinberg et al., 2010*). In contrast, since Na$^+$ channel clustering in the CNS is temporally correlated with assembly of the paranodal junction (*Rasband et al., 1999*), but not the assembly of the CNS nodal ECM (*Susuki et al., 2013*), our results are consistent with the idea that the main driver of CNS nodal Na$^+$ channel clustering is the paranodal cytoskeleton-based mechanism. Since the main paranode-based mechanism is intact in *Six3-Cre;Nfasc$^{fl/fl}$* mice, their Na$^+$ channels may be more stable.

Our discovery that paranodal cytoskeletons play an important role in node of Ranvier formation may also have important implications for disease and injury. Spectrins are potent substrates for calpain-mediated proteolysis (*Siman et al., 1984*). Thus, injury or disease could affect the integrity of this clustering mechanism. This could be particularly detrimental for peripheral and central

autoimmune diseases that have as their molecular targets nodal antigens including NF186 (*Mathey et al., 2007*; *Ng et al., 2012*; *Uncini et al., 2013*) resulting in a 'double-hit': disruption of nodal neurofascin and paranodal junction cytoskeletons. Consistent with this idea, calpain inhibitors preserve nodes of Ranvier in animal models of Guillain-Barre syndrome where nodal antigens are thought to be targeted in the disease (*McGonigal et al., 2010*).

In conclusion, we confirmed that two glia-dependent mechanisms control the clustering of Na$^+$ channels at the nodes of Ranvier. These mechanisms depend on specific cell adhesion molecules that mediate the contact between myelinating glia and their underlying axons at the forming nodes and the paranodal junction. Furthermore, we extended these conclusions by showing that the para-nodal junction-dependent mechanism of Na$^+$ channel clustering requires the $\beta$II spectrin-based cyto-skeleton that is assembled at this site.

## Materials and methods

### Animals

Animals were housed at the Center for Laboratory Animal Care at Baylor College of Medicine, the Weizmann Institute of Science, and the University of Edinburgh. All procedures were approved by the Institutional Animal Care and Use Committees of each institution, and conform to the United States Public Health Service Policy on Human Care and Use of Laboratory Animals. The production of the *Sptbn1*$^{fl/fl}$ (also referred to as *Spnb2*$^{fl/fl}$) and floxed exon 4 *Nfasc*$^{fl/fl}$ mice have been described (*Zonta et al., 2011*; *Zhang et al., 2013*). The floxed exon 6 + 7 *Nfasc*$^{fl/fl}$ mice were generated by the Mouse Clinical Institute – Institut Clinique de la Souris (ICS) in France (mouse line E160). The *Pgk-Cre*, *Avil-Cre* and *Six3-Cre* have all been described elsewhere (*Lallemand et al., 1998*; *Furuta et al., 2000*; *Zhou et al., 2010*). *Thy1-Cre* mice were generated as described for *Thy1-CreERT2* mice (*Zonta et al., 2011*) except the sequence encoding Cre in the *pCreERT2* vector was amplified by PCR to introduce flanking *Xho1* sites which were then used to introduce the Cre sequence into the *pTSC21k* vector. *Caspr*$^{fl/fl}$ mice were generated by a standard gene targeting approach, using a vector containing a neomycin (Neo) resistance gene flanked by two FRT sites, 5' 3.8 kb and 3' 3.2 kb fragments homologous to the genomic *Caspr* locus, and two *loxP* sites (*Golan et al., 2013*). Targeted mice were crossed with FLP deleter mice (*Farley et al., 2000*) to remove the Neo cassette, leaving the first targeted exon floxed by a pair of *loxP* sites.

### Antibodies

We used the following mouse antibodies: anti-Pan Na$^+$ channel (*Rasband et al., 1999*; RRID:AB_477552), anti-ankyrinG (clones N106/36 [RRID: AB_10673030] and 106/65 [RRID: AB_10675130]; Neuromab), anti-Kv1.1 (clone K36/15 [RRID: AB_2128566]; Neuromab), anti-gliomedin (*Eshed et al., 2005*), and anti-$\beta$ tubulin I (Sigma-Aldrich T7816, clone SAP 4G5; RRID: AB_261770). We used the following rabbit antibodies: anti-$\beta$IV spectrin (*Yang et al., 2004*; RRID: AB_2315634), anti-Nav1.6 (Alomone labs, ASC_009; RRID:AB_2040202), anti-Kv1.2 (*Ogawa et al., 2008*) and anti-Neurofascin 186 (MNF2; (*Tait et al., 2000*). We used a sheep pan anti-Neurofascin antibody (PanNF, NFC3) raised against the identical peptide previously described for a rabbit antibody (NFC1; (*Tait et al., 2000*). We used the following chicken antibodies: anti-Pan Neurofascin (AF3235, R&D systems; RRID:AB_10890736). We used the following rat antibodies: anti-myelin basic protein (MBP) (Chemicon, MAB386; RRID:AB_94975). Fluorescent secondary antibodies were purchased from Invitrogen and Jackson Laboratories.

### Immunofluorescence

Optic and peripheral nerves were rapidly dissected and immediately fixed in ice-cold 4% paraformaldehyde in 0.1 M phosphate buffer (PB), pH 7.2, for 30 min. After fixation, nerves were transferred to ice-cold 20% sucrose (w/v) in 0.1 M PB until equilibrated. The tissue was then frozen in Tissue-Tek OCT mounting medium. Teased sciatic nerves were prepared and immunolabeled as previously described (*Poliak et al., 2001*). Briefly, sciatic nerves were fixed as described above, de-sheathed and teased using fine forceps on SuperFrost Plus slides (Menzel-Gläser, Thermo Scientific), air dried over-night, and then kept frozen at −20°C till use. For the isolation of dorsal and ventral roots, spines were dissected and submerged in 2% fresh PFA for 2 hr at 4°C. They were then transferred to

0.5M EDTA, pH 8.0, overnight on ice at 4°C. The next day, spinal cords were excised from the spinal bones with their bound ventral and dorsal roots, and immersed in ice cold PBS overnight on ice at 4°C. The next day, we carefully removed the roots and teased the nerve bundles with fine forceps or thin needles on SuperFrost Plus slides, air dried over-night, and then kept frozen at −20°C till use. In some cases, optic nerves and dorsal roots were sectioned and mounted on slides. All tissues for immunostaining were blocked and permeabilized in 0.1 M PB, 10% normal goat serum, and 0.3% or 0.5% TX-100 for one hour. Primary antibodies were then diluted either in the same blocking buffer or in blocking buffer containing 0.1% TX-100 and added to sections overnight. Labeled sections were then washed 3X in the blocking buffer or PBS for 5 min each. Secondary antibodies were then diluted in blocking buffer and added to the tissue sections for one hour. Sections were then washed three times in PBS for 5 min each. The fluorescently immunolabeled sections were then covered with coverslips using anti-fade mounting medium (KPL, MD) and imaged using a Zeiss Axioimager Z1 with apotome attachment for structured illumination, a Zeiss LSM700 confocal microscope or Leica TCL-SL confocal microscope and 63X objective, numerical aperture 1.4. Images were acquired and processed using the Zen software (Carl Zeiss) or Leica proprietary software. Signal intensity analysis was performed on images obtained from the Zeiss confocal microscope, and performed using Velocity (Perkin-Elmer) or using Zeiss Zen software. For the measurement of node fluorescence intensity, sections were carefully handled side-by-side with the same stainings, exposure times, etc. Nodal intensities were obtained using two separate methods: 1) by dividing the fluorescence intensity by the area of the node, and 2) normalizing the nodal fluorescence intensity to that of the flanking para-nodal Caspr. Both methods were used in the PNS and gave similar results. The data presented in the PNS (as shown in *Figure 1f*) was obtained in the Peles laboratory using the first method, while the data in the CNS was obtained in the Rasband laboratory using the second method.

## Electrophysiology

CAP recordings in dorsal roots and optic nerves were performed as described elsewhere (*Rasband et al., 1999*; *Susuki et al., 2007*). Briefly, roots were rapidly dissected and placed in a continuously perfused, oxygenated, and temperature controlled (23°C) recording chamber. A standard Locke's solution consisting of (in mM): NaCl 154, KCl 5.6, CaCl2 2,d-glucose 5, and HEPES 10, pH 7.4. The ends of each root was drawn into a suction electrode and after stimulation responses were recorded. Stimuli consisted of 50 μsec pulses with amplitudes adjusted to ~10% above the level required for a maximum response. Conduction velocities were calculated by measuring the length of the nerve and dividing this by the latency from stimulation to the peak of the CAP.

## Electron microscopy

Electron microscopy was performed as described before (*Chang et al., 2010*). Wild type and mutant animals were sacrificed, and their sciatic nerves exposed and fixed by continuous dripping of fresh fixative for 40 min (fixative containing 4% paraformaldehyde, 0.1 M sodium cacodylate, and 2.5% glutaraldehyde, pH 7.4, in PBS). Sciatic nerves were then carefully dissected, placed in fresh fixative, and left to rotate over-night at ambient temperature while protected from light. Samples were then transferred to 4°C till processing. For the fixation of dorsal and ventral roots, spines were dissected and submerged in fixative, rotating over-night at ambient temperature while protected from light. The next day dorsal and ventral roots were very carefully dissected and placed in fresh fixative, and then kept at 4°C till processing. Processing was carried out as previously described (*Poliak et al., 2003*). Briefly, tissues were washed four times in 0.1 M cacodylate-buffer, incubated with gentle agitation for 1 hr in osmium solution (1% OsO4, 0.5% K2Cr2O7 and 0.5% K4[Fe(CN)6]·3H2O in 0.1 M cacodylate-buffer) and subsequently washed with 0.1 M cacodylate-buffer and with millipore filtered H2O. For better contrast resolution, tissues were impregnated with uranyl-acetate (2% in H2O for 1 hr) before being washed twice in H2O. After a series of dehydrating steps in rising EtOH-concentrations (50%, 70%, 96% each twice for 5 min, 100% three times 10 min), tissues were embedded in propylene oxide (3 × 10 min) and incubated overnight in 30% Epon 'Hard' (EMS) diluted in propylene oxide. Next, Epon was introduced gradually into the samples (50% and 70% 24 hr each, 100% 48 hr). Sciatic and optic nerves were then transferred into backing-molds with 100% Epon and left to harden for three days in an oven at 65°C. Sections were subsequently sectioned and examined using a Philips CM-12 transmission electron microscope.

## Tissue lysate preparation and Western blot analysis

Dissected tissues were snap-frozen in liquid nitrogen and kept at −80°C until processing. Homogenization was performed in RIPA buffer (50 mM Tris-HCl pH = 7.4, 1% NP-40, 0.25% Sodium-deoxycholate, 150 mM NaCl and 1 mM EDTA in H2O, supplemented with a protease inhibitors cocktail (Sigma-Aldrich). Homogenates were incubated on ice for 40 min (vortexed every 5 min), centrifuged at 15,000 rpm for 30 min and supernatant was collected and kept at −80°C. Protein concentration was determined using a BCA kit (Pierce). Lysates were resolved by SDS-PAGE in Tris-Acetate 7% gels, and transferred to a nitrocellulose membrane (110V, 1 hr). Membranes were then incubated with blocking solution (5% BSA in TBS-T (5 mM Tris, 15 mM NaCl and 0.05% tween in H2O pH = 7.5)) for 60 min and reacted with the appropriate antibodies for 12 hr at 4°C. Membranes were then washed in TBS-T (3 × 5 min) and incubated with horseradish-peroxidase-coupled secondary antibody for 45 min at room temperature. Membranes were washed again and reacted with ECL (Pierce). Membranes were visualized using the Chemidoc MP Digital chemiluminescent gel documentation system (Bio-Rad).

## Dissociated dorsal root ganglia (DRG) myelinating cultures

Dissociated DRG mixed myelinating cultures were prepared as described (*Eshed et al., 2005*). Dorsal root ganglia were dissected from mouse embryos at 13.5 days of gestation, collected in Leibovitz's L15 medium (Gibco) and dissociated with trypsin (without EDTA; Cat. # 25050, Gibco). After dissociation, 40,000 cells were plated on EtOH-washed 13 mm glass-coverslips (Thermo) pre-coated with 0.4 µg/ml Matrigel (BD Biosciences) and 10 µg/ml Poly-D-lysine (Sigma-Aldrich). DRG cultures were maintained for a day in NB medium (2% B27 (Gibco), 50 ng/ml NGF (Alomone labs) and 1% glutamax in neurobasal medium (Gibco)) and then switched to BN medium (1% ITS supplements (Sigma- Aldrich), 0.2% BSA, 4 g/L D-glucose, 50 ng/ml NGF and 1% glutamax in basal Eagle's medium (BME; Gibco)). Myelination was induced after 10 additional days by supplementing with 50 µg/ml L-ascorbic acid (Sigma-Aldrich) and 15% heat inactivated FCS (BNC medium).

## Statistical analysis

Sets of age-matched cKO or KO mice of mixed sexes and their controls were randomly collected from littermates or from litters which had similar dates of birth. Experimenters were blinded to the genotype of the animals. The statistical significance of each comparison was determined using a student's t-test or one-way ANOVA. Results are given as mean +/− SEM.

## Acknowledgements

This work was supported by NIH grants (NS069688 and NS044916 to MNR, and NS50220 to EP), the Dr. Miriam and Sheldon Adelson Medical Research Foundation (MNR and EP), the US-Israel Binational Science Foundation (MNR and EP), The Israel Science Foundation (EP), the Wellcome Trust (PJB) and MRC (PJB). EP is the Incumbent of the Hanna Hertz Professorial Chair for Multiple Sclerosis and Neuroscience.

## Additional information

### Funding

| Funder | Grant reference number | Author |
|---|---|---|
| Wellcome | | Peter J Brophy |
| Medical Research Council | | Peter J Brophy |
| National Institutes of Health | NS044916 | Matthew N Rasband |
| Dr. Miriam and Sheldon G. Adelson Medical Research Foundation | | Matthew N Rasband Elior Peles |
| United States-Israel Binational Science Foundation | | Matthew N Rasband Elior Peles |

| National Institutes of Health | NS069688 | Matthew N Rasband |
|---|---|---|
| National Institutes of Health | NS050220 | Elior Peles |
| Israel Science Foundation | | Elior Peles |

The funders had no role in study design, data collection and interpretation, or the decision to submit the work for publication.

**Author contributions**

VA, CZ, AV, AZ, DRZ, YE-E, Data curation, Formal analysis, Investigation; PJB, Conceptualization, Formal analysis, Supervision, Funding acquisition, Methodology, Project administration, Writing—review and editing; MNR, EP, Conceptualization, Formal analysis, Supervision, Funding acquisition, Validation, Investigation, Methodology, Writing—original draft, Project administration, Writing—review and editing

**Author ORCIDs**

Matthew N Rasband, http://orcid.org/0000-0001-8184-2477

**Ethics**

Animal experimentation: Animals were housed at the Center for Laboratory Animal Care at Baylor College of Medicine (protocol AN4634), the Weizmann Institute of Science, and the University of Edinburgh. All procedures were approved by the Institutional Animal Care and Use Committees of each institution, and conform to the United States Public Health Service Policy on Human Care and Use of Laboratory Animals.

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
