## [Decision Letter]

Thank you for submitting your article "The Paranodal Cytoskeleton Clusters Na^+^ Channels At Nodes of Ranvier" for consideration by *eLife*. Your article has been reviewed by three peer reviewers, one of whom is a member of our Board of Reviewing Editors, and the evaluation has been overseen by Richard Aldrich as the Senior Editor.

The reviewers have discussed the reviews with one another and the Reviewing Editor has drafted this decision to help you prepare a revised submission.

This study re-evaluates the role of nodal versus paranodal interactions in clustering sodium channels (NaChs) at nodes of Ranvier in the PNS and CNS. Using in vivo genetic manipulations to selectively delete NF186 from different populations of PNS and CNS neurons, the authors find NaV channel clusters remain at nodes in the absence of NF186. This finding is, on the surface, different from that reported by Thaxton et al. in 2011, in which loss of NF186 was reported to result in a loss of NaCh clusters. The authors suggest that this discrepancy is due to the poor efficiency of the Cre line used in the 2011 report, and further, that NF186 plays an important role in maintaining NaCh clusters rather than in initially forming clusters. Consistent with this interpretation, the 2011 report observed some NaCh clusters in P11 sciatic nerves, and both groups find a decrease in NaCh clusters with age.

This study is important, as it provides a compelling explanation for previous findings, reconciles data from different manipulations, and describes a role for glial-axonal paranodal interactions through NF155 and βII spectrin in the initial localization of NaV channels to nodes. The in vivo analyses and use of multiple cre lines enhance the findings and impact. There was a consensus that the manuscript is appropriate for publication in *eLife*, provided the following concerns are addressed.

Concerns with the text:

1) In some positions, the paper is written in a way as if almost nothing was known before. For example, Abstract: "but the specific mechanisms remain unclear." It would be important to carefully re-write the paper to make clear where the model established for example in Susuki et al. has been reinforced and where it has been extended. Novel aspects established in their current study should be highlighted, for example, the role of paranodal spectrin and the role of neurofascin186 in maintaining sodium channel clustering.

2) The paper would benefit from a graphical model depicting the key insights of the paper and a model that summarizes our current understanding of node assembly.

3) It isn't clear from the experiments whether the pathway proposed is merely an example of redundancy or whether this interaction is important for some aspect of initial node formation. It seems that βII spectrin deletion alone does not alter nodal clustering, suggesting that under normal conditions, NF186 is sufficient. It would be helpful if the authors were to discuss this issue and provide some quantification of NaCh density in the βII knockout (see below).

4) Some of the PNS experiments deleting NF186 appear to repeat work that is already published. Thus, Feinberg et al., 2010, cocultured DRG neurons from NF-/- mice with wt Schwann cells and found Na^+^ channels clustered at nodes, but not heminodes. This is also reported here in Figure 1, although with a different mutant. Was a different result expected? The Salzer lab used RNAi to knock down NF186 in similar cultures, and found no Na^+^ channel clustering at regions that they interpreted as nodes, but had wide gaps (JCB 177:857, 2007).

Concerns about immunocytochemical quantification:

1) Results section. NaCh immunofluorescence intensity was measured in the conditional deletion and compared with a control. This is a very difficult measurement, requiring careful controls. All we are told is that they used the Volocity software. For example, comparing the dorsal root images in Figure 1, the nodal label looks weaker in the deletion, but so does the Caspr label. How were the measurements normalized to be able to compare different sections? Were the sections mounted on the same slide so that they would be treated identically?

2) In the Abstract and the Discussion the authors do not address the fact that NaCh clustering, while present in the absence of NF186, is markedly decreased and the mice have a profound deficit in action potential conduction. Thus, the results provide a more nuanced picture than is presented. It would be helpful if this issue were addressed. As the density of NaCh is affected, the authors should provide a consistent assessment of NaCh content in the different mouse lines (Although immunolabeling is not truly quantitative, it would provide a means to compare relative abundance, provided processing and image acquisition are standardized). For example, in Figure 1, NaCh immunostaining intensity is quantified, providing an explanation for the reduction in AP conduction velocity, but similar data are not provided for the other mouse lines and no density measures are provided for CNS axons.

3) The authors conclude that paranodal interactions are "sufficient for the initial NaCh clustering at PNS nodes." However, they do not provide evidence that the density of NaChs achieved at nodes in NF186 deficient axons is similar to that which occurs in wild type. This statement should be revised or additional data provided to support this conclusion.

4) As pointed out by the authors, some nodes in *Six3-Cre;Nfasc^fl/fl^*mice at P60 lacked NaCh clusters but had paranodal NF immunoreactivity. The authors need to assess whether these axons had intact βII spectrin. If so, it would contradict the primary conclusion of the study and should be discussed. Is it possible that different neurons may exhibit a different dependence on NF186/NF155 interactions for NaCh clustering and/or maintenance?

5) In Figure 1—figure supplement 2 and Figure 2—figure supplement 1, it would be helpful if the authors provided quantification of the number of positive nodes with βII spectrin, βIV spectrin, AnkG, etc. (Figure 1—figure supplement 2) to demonstrate the robustness of these observations. 2. Figure 4 and subsection “The spectrin-based paranodal cytoskeleton underlies the paranodal clustering mechanism”. The thinner myelin seen in NF186/βII spectrin null mice is interesting. The authors do not speculate on a possible mechanism for this. Further, while they state that paranodal junctions are intact, no transverse bands are visible in Figure 4, while the control (Figure 4) has hints of them. The presence of Caspr may not guarantee "intact paranodal junctions". The authors may wish to refer to Mierzwa et al., J. Comp. Neurol. 518:2841, 2010.

Concerns about the electrophysiological analysis

1) The electrophysiology raises a number of questions. Firstly, the methods simply refer to previous papers. There should be a few sentences describing: the basic technique (wires? suction electrodes?); stimulus parameters; the bath solution recipe and if it was oxygenated; and the temperature. Secondly, it would be helpful to have a conduction velocity for all peaks. Is the second peak in Figure 1—figure supplement 1 attributable to unmyelinated axons? Why is there such a large range of velocities in the control in Figure 1—figure supplement 1? Some of these may be so slow as to be from unmyelinated fibers. The *Avil-Cre* NF(f/f) in Figure 1—figure supplement 1 does not just show a decreased velocity of the fast peak. There is a very pronounced second peak. What is its velocity? Does it represent small myelinated axons or unmyelinated axons? Has there been a shift in the relative numbers of these fibers from the NF(f/f) control?

2) Could the decreased conduction velocity in the CNS in the *Six3-Cre* NF(f/f) optic nerves be due to the loss of ECM proteins that was seen? The authors have shown previously that loss of Tenascin-R lowers velocity.

---

## [Author Response]

*Concerns with the text:*

*1) In some positions, the paper is written in a way as if almost nothing was known before. For example, Abstract: "but the specific mechanisms remain unclear." It would be important to carefully re-write the paper to make clear where the model established for example in Susuki et al. has been reinforced and where it has been extended. Novel aspects established in their current study should be highlighted, for example, the role of paranodal spectrin and the role of neurofascin186 in maintaining sodium channel clustering.*

We have carefully revised our manuscript to make emphasize what was has been reinforced or confirmed and what is novel about our study. We carefully examined every major section (see summary below):

Abstract: deleted statement that the "[…] specific mechanisms remain unclear."

Introduction: we state what previous evidence supported a paranodal junction-dependent mechanism, and we highlight where the controversy remains surrounding the necessity of NF186. Finally, we conclude the introduction by stating that we confirmed the role of the paranodal junction and extend our understanding of node formation by revealing the requirement for the βII spectrin-dependent paranodal cytoskeleton.

Results: In the first section that describes the deletion of NF186 and the sufficiency of NF155-dependent paranodal junctions to cluster Na^+^ channels in the PNS we conclude: “Hence, while these observations confirm that paranodal junctions are sufficient for the initial Na^+^ channel clustering at PNS nodes, they also support previous observations that NF186 plays important roles in maintaining the nodal protein complex which is necessary for proper action potential conduction (Amor et al., 2014; Desmazieres et al., 2014).” Throughout the remainder of the results we indicated where the results either confirm previous work or extend our understanding of the mechanisms.

Discussion: We revised the discussion to indicate where we confirm or extend our understanding of node formation. The discussion focuses on three things: 1) the model for how Na^+^ channels are clustered and what the genetic mouse models reveal from the work reported here, 2) that NF186 is dispensable and we provide a potential explanation for why the work of Thaxton et al. differs from our study, and 3) the majority of the discussion focuses on the paranodal cytoskeleton and its role in Na^+^ channel clustering which is the major conceptual advance reported here.

*2) The paper would benefit from a graphical model depicting the key insights of the paper and a model that summarizes our current understanding of node assembly.*

We now include a model (new Figure 5.) that summarizes the major conclusions from the paper in the Discussion section.

*3) It isn't clear from the experiments whether the pathway proposed is merely an example of redundancy or whether this interaction is important for some aspect of initial node formation.*

We agree that we have not provided any evidence that the paranodal mechanism plays an important role in some aspect of initial node formation. In the PNS the primary mechanism of NaCh clustering is through Gldn/NF186 with the paranodal junctions being the secondary mechanism. In the CNS, the primary mechanism is the paranodal junction, while the ECM/NF186 interaction is the secondary mechanism. These conclusions were reported by us previously in Susuki et al., Neuron 2013.

*It seems that βII spectrin deletion alone does not alter nodal clustering, suggesting that under normal conditions, NF186 is sufficient. It would be helpful if the authors were to discuss this issue and provide some quantification of NaCh density in the βII knockout (see below).*

The reviewer is correct that there is no difference in Na^+^ channel clustering in the βII spectrin cKO mice. We performed new measurements of nodal densities and found that 94.1% and 94.6% of nodes in control and *Avil-Cre;Spnb2^fl/fl^*mice, respectively, had Na^+^ channels (p=0.28 by Student’s t-test n=3 independent animals for each genotype and a total of 796 and 464 nodes were examined in control and cKO mice, respectively). When we examined the Na^+^ channel density at individual nodes in the *Avil-Cre;Spnb2^fl/fl^*mice we found fluorescence intensities of 1.035 and 0.935 in control and *Avil-Cre;Spnb2^fl/fl^*mice, respectively (p=0.44 by Student’s t-test, n= 4 mice of each genotype with a total of 100 and 92 nodes measured in control and cKO mice, respectively). We have added these details in the Results section.

*4) Some of the PNS experiments deleting NF186 appear to repeat work that is already published. Thus, Feinberg et al., 2010, cocultured DRG neurons from NF-/- mice with wt Schwann cells and found Na^+^ channels clustered at nodes, but not heminodes. This is also reported here in Figure 1, although with a different mutant. Was a different result expected? The Salzer lab used RNAi to knock down NF186 in similar cultures, and found no Na^+^ channel clustering at regions that they interpreted as nodes, but had wide gaps (JCB 177:857, 2007).*

The reviewer is correct that the conclusion from the experiment reported in Figure 1 is similar to that reported in Feinberg et al. (2010) and that reported by Dzhashiashvili et al. (2007). A different result was not expected. The purpose of the figure was to show that using the new conditional NF186 knockout mice we can confirm previous studies using mixed cultures. Again these data are in support of a model where NF186 is dispensable for nodal clustering, but required for heminodal clustering of channels. Given previous reports to the contrary (Thaxton et al., 2011) we consider these experiments to be important for the main messages.

*Concerns about immunocytochemical quantification:*

*1) Results section. NaCh immunofluorescence intensity was measured in the conditional deletion and compared with a control. This is a very difficult measurement, requiring careful controls. All we are told is that they used the Volocity software. For example, comparing the dorsal root images in Figure 1, the nodal label looks weaker in the deletion, but so does the Caspr label. How were the measurements normalized to be able to compare different sections? Were the sections mounted on the same slide so that they would be treated identically?*

For the measurement of fluorescence intensity, sections were carefully handled side-by-side with the same stainings, exposure times, etc. Nodal intensities were obtained by dividing the fluorescence intensity by the area of the node (for the data shown in Figure 1). Nevertheless, to be absolutely certain of our results we performed two independent measurements of NaCh density in the *Avil-Cre;Nfasc^fl/fl^*mice – one in the Peles lab and one in the Rasband lab. The measurements shown in Figure 1 are from the Peles lab. The Rasband lab measured nodal Na^+^ channel intensity at P10, but normalized the intensity to that for the flanking paranodal Caspr. We obtained a ratio of 0.94 in *Nfasc^fl/fl^*mice mice and 0.74 in *Avil-Cre;Nfasc^fl/fl^*mice (p=0.02, Student’s t-tets, n= 4 mice of each genotype). Thus, using a different method of measuring nodal Na^+^ channel intensity both groups obtained consistent results. We chose to present the data from the Peles lab in the manuscript because we had two time points.

*2) In the Abstract and the Discussion the authors do not address the fact that NaCh clustering, while present in the absence of NF186, is markedly decreased and the mice have a profound deficit in action potential conduction. Thus, the results provide a more nuanced picture than is presented. It would be helpful if this issue were addressed. As the density of NaCh is affected, the authors should provide a consistent assessment of NaCh content in the different mouse lines (Although immunolabeling is not truly quantitative, it would provide a means to compare relative abundance, provided processing and image acquisition are standardized). For example, in Figure 1, NaCh immunostaining intensity is quantified, providing an explanation for the reduction in AP conduction velocity, but similar data are not provided for the other mouse lines and no density measures are provided for CNS axons.*

To be as clear as possible about the major conclusions of our study we focused our Abstract on the three major results: 1) NF186 is dispensable for Na^+^ channel clustering, 2) paranodes are sufficient to cluster Na^+^ channels, and 3) the molecular basis for the paranodal mechanism is βII spectrin. We do not believe the Abstract is an appropriate place for a detailed description of results. However, we have not suppressed or hidden the observation of reduced Na^+^ channel density. We agree the reduction in Na^+^ channel density is significant and we report this in Figure 1. We now provide fluorescence intensity measurements for the βII spectrin conditional knockout mouse line in the revised manuscript, but we do not have fluorescence intensity measurements for the *Thy-1Cre;Nfasc(4)^fl/fl^* mice. We also performed fluorescence intensity measurements of Na^+^ channel intensity in the *Six3Cre;Nfasc^fl/fl^*CNS.

Surprisingly, in contrast to the PNS where there was a very obvious reduction in NaCh density at individual nodes (Figure 1), we measured no equivalent change in the *Six3Cre;Nfasc^fl/fl^*CNS optic nerve (n= 3 animals of each genotype). These data are included here for the reviewers Figure 6. We speculate that this discrepancy between the CNS and PNS reflects differences in primary and secondary clustering mechanisms in the PNS vs. the CNS. Specifically, in the PNS the primary mechanism of NaCh clustering is through Gldn/NF186 with the paranodal junctions being the secondary mechanism. In the CNS, the primary mechanism is the paranodal junction, while the ECM/NF186 interaction is the secondary mechanism (Susuki et al., Neuron 2013). (These conclusions are also consistent with the observation that in mice lacking paranodal junctions the nodal clusters in the CNS become proportionally much, much longer than those in the PNS (Rios et al., J Neurosci 2003)). We chose not to include these data since they do not change the conclusions and would require a complicated discussion of the differences in the hierarchy of clustering mechanisms between the PNS and CNS that may detract from the main message. Should the reviewers feel these data are critical to the overall message of the paper we will be happy to include them as they provide strong support for the sufficiency of the paranodal mechanism – especially in the CNS. Finally, in recognition of the reviewer’s point that our results are more nuanced (i.e. that clustering is not completely normal since nodes have reduced densities of Na^+^ channels), we revised the Discussion and overall conclusion to state, “Accordingly, the loss of NF186 can be partially compensated for by the paranodal junction-based cytoskeletal barrier since Na^+^ channels may still be clustered, but they cannot be maintained in the absence of NF186 (Figure 5).”

Author response image 1.Nodal NaCh intensity in the *Six3Cre;Nfasc^fl/fl^*CNS optic nerve.**DOI:**
http://dx.doi.org/10.7554/eLife.21392.011

*3) The authors conclude that paranodal interactions are "sufficient for the initial NaCh clustering at PNS nodes." However, they do not provide evidence that the density of NaChs achieved at nodes in NF186 deficient axons is similar to that which occurs in wild type. This statement should be revised or additional data provided to support this conclusion.*

The results presented in this paper, using 3 different conditional knockout mouse models, show that paranodal junctions are sufficient for the initial clustering of Na^+^ channels (Figure 1, Figure 2, and Figure 1—figure supplement 2). We don’t claim that the density of channels is identical to that of control animals. We specifically only stated that paranodes are sufficient to induce Na^+^ channel clustering at PNS nodes. In fact, we provided data in Figure 6, suggesting that in contrast to the PNS, CNS nodes may have normal densities of Na^+^ channels even at P17, which is the time of peak node formation in the optic nerve (see Rasband et al., J Neurosci 1999). For the reasons already stated (see above), we chose not to include these results. We conclude that indeed paranodes are sufficient for the initial Na^+^ channel clustering at PNS nodes. Please note the model shown in Figure 5 shows reduced Na^+^ channel densities in NF186-deficient mice, emphasizing that the paranodal clustering mechanisms may not be sufficient to maintain the normal density of channels.

*4) As pointed out by the authors, some nodes in Six3-Cre;Nfasc^fl/fl^ mice at P60 lacked NaCh clusters but had paranodal NF immunoreactivity. The authors need to assess whether these axons had intact βII spectrin. If so, it would contradict the primary conclusion of the study and should be discussed. Is it possible that different neurons may exhibit a different dependence on NF186/NF155 interactions for NaCh clustering and/or maintenance?*

The reviewer suggests that if we were able to identify a P60 node that lacked Na^+^ channels but still had intact βII spectrin it would argue against a model where the paranodal cytoskeleton is sufficient for channel clustering. In fact, we expect that all P60 nodes in the *Six3-Cre;Nfasc^fl/fl^*mice have intact paranodal βII spectrin. Since earlier timepoints always show clustered Na^+^ Channels (Figure 2), we conclude that loss of NF186 results in failure to maintain the clusters of Na^+^ channels, not to assemble them. If we detected paranodal βII spectrin at P60 flanking nodes lacking Na^+^ channels in these mice it would not argue against our model. Unfortunately direct demonstration of βII spectrin at CNS paranodes in the *Six3-Cre;Nfasc^fl/fl^*miceis technically exceedingly difficult (nigh impossible for the number of instances that would be needed to be observed for statistical confidence) for two reasons: 1) the frequency of a node lacking Na^+^ channels is very rare (only ~15% of nodes at P60), and 2) βII spectrin is highly expressed in both axons and myelinating oligodendrocytes making it very difficult to observe paranodal βII spectrin in the CNS. In addition, the density of myelinated axons in the optic nerve makes the detection of paranodal βII spectrin very difficult. It is only in regions where optic nerve axons have ‘frayed’ and individual axons can be seen where it is possible to see the paranodal βII spectrin. It is much easier to see the paranodal βII spectrin in PNS axons because they can be much more easily separated.

*5) In Figure 1—figure supplement 2 and Figure 2—figure supplement 1, it would be helpful if the authors provided quantification of the number of positive nodes with βII spectrin, βIV spectrin, AnkG, etc. (Figure 1—figure supplement 2) to demonstrate the robustness of these observations.*

We are unsure of the reviewers’ intent with this question. Is the suggestion that Na^+^ channels may be clustered in the absence of βIV spectrin or AnkG? We previously showed that the ankG-binding motif of Na^+^ channels is both necessary and sufficient for the proper clustering of nodal Na^+^ channels (Gasser et al., J Neurosci 2012) and we showed that Na^+^ channels are never clustered without a nodal ankyrin (Ho et al., Nat Neurosci 2014). Based on these prior observations we did not quantify βIV spectrin or ankG staining in the mice we analyzed – instead our focus was on Na^+^ channels since clustering of Na^+^ channels requires intact ankyrin and β spectrin clustering. Nevertheless, our qualitative analysis indicated that the staining for ankG and βIV spectrin in *Avil-Cre;Nfasc^fl/fl^;Spnb2^fl/fl^* mice is similar to that for Na^+^ channels. Since the animals analyzed in Figure 1—figure supplement 2 (*Thy-1Cre;Nfasc(4)^fl/fl^* mice) have intact βII spectrin gene expression and intact paranodal junctions (as indicated by Caspr staining) we did not analyze βII spectrin staining in these mice. There is no reason to expect that βII spectrin staining should be altered.

*Figure 4 and subsection “The spectrin-based paranodal cytoskeleton underlies the paranodal clustering mechanism”. The thinner myelin seen in NF186/βII spectrin null mice is interesting. The authors do not speculate on a possible mechanism for this. Further, while they state that paranodal junctions are intact, no transverse bands are visible in Figure 4, while the control (Figure 4) has hints of them. The presence of Caspr may not guarantee "intact paranodal junctions". The authors may wish to refer to Mierzwa et al., J. Comp. Neurol. 518:2841, 2010.*

We agree that the thinner myelin observed in the *Avil-Cre;Nfasc^fl/fl^;Spnb2^fl/fl^* is quite interesting and also consistent with a delay in myelination observed in the full NFasc knockout mouse (Zonta et al., JCB 2008). However, we chose not to discuss the hypomyelination because the focus of the paper is on Na^+^ channel clustering at nodes and the function of the paranodal cytoskeleton in that process. Although interesting, we believe it is tangential to the main message of the paper and we have no data to support any particular mechanism. If the editor would like us to discuss the issue we are happy to do it, but it will be speculation. The more important question is about the EM and the lack of transverse bands. We agree that transverse bands are not visible in the example of the paranode shown. Unfortunately, because of the recombination that occasionally occurs in oligodendrocytes (see Figure 2—figure supplement 1), we cannot be certain that a given paranode has NF155. Although most investigators believe the molecular basis of the transverse band is the Caspr/contactin/NF155 protein complex, this has never been proven by immunoelectron microscopy. Nevertheless, since there are no known examples where transverse bands are present but Caspr and NF155 are not also clustered, we used Caspr and paranodal NF immunostaining to indicate intact paranodal junctions in the nodes we analyzed. Thus, although we are limited in our analysis of paranodes by EM, the immunostaining strongly supports the conclusion that junctions are intact.

*Concerns about the electrophysiological analysis*

*1) The electrophysiology raises a number of questions. Firstly, the methods simply refer to previous papers. There should be a few sentences describing: the basic technique (wires?, suction electrodes?); stimulus parameters; the bath solution recipe and if it was oxygenated; and the temperature.*

We have added these details to the Materials and methods section.

*Secondly, it would be helpful to have a conduction velocity for all peaks. Is the second peak in Figure 1—figure supplement 1 attributable to unmyelinated axons?*

The second peak in Figure 1—figure supplement 1 is unlikely to reflect an increase in unmyelinated axons since their conduction velocities and amplitudes are more than an order of magnitude lower than signals recorded from myelinated sensory axons, and we see no reduction in myelination in the *Avil-Cre;Nfasc^fl/fl^* or *Six3-Cre;Nfasc^fl/fl^*mice. Instead the delay in the first peak and increase in the second peak corresponds to slowing of axons due to reduced Na^+^ channel densities. Respectfully, we do not believe reporting a conduction velocity for all peaks is particularly informative since the entire trace shown represents the summation of action potentials from many hundreds if not thousands of axons in the sensory roots. The velocity for the second peak is easy to provide, it is just that we do not think it will provide any additional information beyond what is already given in Figure 1—figure supplement 1. Perhaps we have overlooked the reason why the reviewer suggested this. If the reviewer feels strongly about this we would be happy to consider their reasons.

*Why is there such a large range of velocities in the control in Figure 1—figure supplement 1? Some of these may be so slow as to be from unmyelinated fibers. The Avil-Cre NF(f/f) in Figure 1—figure supplement 1 does not just show a decreased velocity of the fast peak. There is a very pronounced second peak. What is its velocity? Does it represent small myelinated axons or unmyelinated axons? Has there been a shift in the relative numbers of these fibers from the NF(f/f) control?*

The reason for the large range of velocities in the control in Figure 1—figure supplement 1 is that we pooled all CAP data recorded from dorsal roots. These roots have different proportions of sensory neuron subtypes with different diameters which results in a wide distribution of conduction velocities even in the control roots. Please see the answer above with regard to the unmyelinated axons and the second peak.

2) Could the decreased conduction velocity in the CNS in the Six3-Cre NF(f/f) optic nerves be due to the loss of ECM proteins that was seen? The authors have shown previously that loss of Tenascin-R lowers velocity.

The reviewer is correct that we (and others) have previously reported changes in CNS nerve conduction velocity in mice lacking nodal ECM proteins such as Tenascin and Bral1 (loss of Bral1 also disrupts Brevican and Versican retention at CNS nodes). Thus, it is quite possible that the loss of these ECM proteins also contributes to reduced conduction velocities. We thank the reviewer for this suggestion and have added the following sentence to the Results section: “The reduced conduction velocity does not reflect impaired myelination (Figure 2—figure supplement 1), but instead likely reflects the combined effects of decreased nodal Na^+^ channel density, loss of nodal extracellular matrix molecules (Weber et al., 1999; Bekku et al., 2010), and loss of some paranodal junctions due to NF155-deficient oligodendrocytes (Figure 2—figure supplement 1).”